# Accelerated Generation of Extra-Islet Insulin-Producing Cells in Diabetic Rats, Treated with Sodium Phthalhydrazide

**DOI:** 10.3390/ijms23084286

**Published:** 2022-04-13

**Authors:** Musa T. Abidov, Ksenia V. Sokolova, Irina F. Gette, Irina G. Danilova

**Affiliations:** 1Institute of Immunopathology and Preventive Medicine, 1000 Ljubljana, Slovenia; alina144@mail.ru; 2Institute of Immunology and Physiology, Russian Academy of Sciences, Pervomajskaya 106, 620049 Yekaterinburg, Russia; i.goette@yandex.ru (I.F.G.); ig-danilova@yandex.ru (I.G.D.)

**Keywords:** extra-islet insulin-producing cell, macrophage, macrophage plasticity, immunomodulators, sodium aminophthalhydrazide, streptozotocin-nicotinamide-induced diabetes, type 2 diabetes mellitus

## Abstract

β-cells dysfunction plays an important role in the pathogenesis of type 2 diabetes (T2D), partially may be compensated by the generation of extra-islet insulin-producing cells (IPCs) in pancreatic acini and ducts. Pdx1 expression and inflammatory level are suggested to be involved in the generation of extra-islet IPCs, but the exact reasons and mechanisms of it are unclear. Macrophages are key inflammatory mediators in T2D. We studied changes in mass and characteristics of extra-islet IPCs in rats with a streptozotocin-nicotinamide model of T2D and after i.m. administration of 20 daily doses of 2 mg/kg b.w. sodium aminophthalhydrazide (APH). Previously, we found that APH modulates macrophage production and increases the proliferative activity of pancreatic β-cells. Expressions of insulin and Pdx1, as well as F4/80 (macrophage marker), were detected at the protein level by immunohistochemistry analysis, the concentration of pro- and anti-inflammatory cytokines in blood and pancreas—by ELISA. Diabetic rats treated with APH showed an increasing mass of extra-islet IPCs and the content of insulin in them. The presence of Pdx1^+^ cells in the exocrine pancreas also increased. F4/80^+^ cell reduction was accompanied by increasing TGF-β1 content. Interestingly, during the development of diabetes, the mass of β-cells decreased faster than the mass of extra-islet IPCs, and extra-islet IPCs reacted to experimental T2D differently depending on their acinar or ductal location.

## 1. Introduction

Type 2 diabetes mellitus (T2D) is a chronic disorder leading to increased rates of micro- and macrovascular complications, metabolic decompensation, and premature mortality. It is estimated that the number of people with diabetes, more than 90% of whom are patients with T2D, will increase to 592 million by 2035 [1], making T2D a global challenge [2]. Prevalence of T2D highlights the urgency for discovering novel approaches to T2D pathogenesis-based therapy. Pancreatic damage is an integral part of the pathogenesis of T2D. The exocrine pancreas is composed of acini, secreting digestive enzymes, and associated ducts, and the endocrine part of the pancreas is represented by the islets of Langerhans, which include β-cells, producing insulin. Over the last decades, much data have been accumulated that emphasize the importance of impaired β-cell function in the pathogenesis of T2D (reviewed in the work of [3]). Moreover, some investigators consider the decrease in morphological mass and functional reserve of β-cells as one of the main mechanisms of the development of T2D [4,5].

The ability to produce insulin is not a privilege only of islet β-cells. Acinar and ductal epithelium may also be present in insulin-positive cells containing granules of insulin or insulin-producing cells (IPCs) [6,7,8]. Non-diabetic healthy mammals’ extra-islet IPCs constitute from 3% [9] to 15% [10] of all IPCs in the pancreas. Considering the fundamental importance of insulin in both physiologically normal and pathological conditions [11,12,13], among which one of the main is diabetes mellitus, it is not surprising that the phenomenon of insulin synthesis in structures not adapted for this in the traditional sense is a keen area of interest. However, sources, mechanisms, and conditions for the generation of extra-islet IPCs remain poorly understood. The recipient of the insulin they produce is not known: does insulin enter the general bloodstream and help to maintain normoglycemia, or does it act para- or autocrine, influencing the processes of differentiation and function of surrounding tissues?

Nevertheless, most authors describing IPCs located in the pancreas outside the islets of Langerhans consider them as a morphological and histological basis for restoring the insulin-producing function of the pancreas in T1D and T2D. A number of studies have demonstrated the amelioration of hyperglycemia in diabetes after an increase in the quantity of extra-islet IPCs [14,15]. The number of GLUT2 receptors on the membrane of extra-islet IPCs is reduced [6], which suggests that they suffer less from hyperglycemia than islet β-cells. Meanwhile, there is clearly insufficient data to characterize the changes occurring with extra-islet IPCs in T2D.

Expression of pancreatic transcription factors, inflammation, and hyperglycemia are among the factors that are supposed to be involved in the origin of extra-islet IPCs. Pancreatic transcription factors (pTF) Pdx1, Ngn3, and MafA are the main regulators for the generation of IPCs, which is confirmed by the transdifferentiation of acinar and ductal pancreatic cells in IPCs by increasing their expression [16,17,18]. Pancreatic and duodenal homeobox factor-1 (Pdx1) plays a crucial role in pancreas formation in both embryonic and postnatal development [19,20], as well as necessary for β-cell maturation and functioning [21,22]. Studies demonstrated that epigenetic changes of the *Pdx1* gene, including both increased DNA methylation and histone modifications, can result in reduced Pdx1 expression and diabetes in postnatal life [23]. Pancreatic islets of patients with T2D demonstrated increased methylation in CpG sites of the distal *Pdx1* promoter and its enhancer and a decrease in *Pdx1* mRNA expression [24]. Promoting Pdx1 expression is suggested to be an effective strategy to preserve the mass and function of IPCs, both intra- and extra-islet. For instance, Yao et al. demonstrated that the natural compound tectorigenin can enhance the activity of the promotor for the *Pdx1* gene via activating extracellular signal-related kinase, and this effect was observed under either normal or glucotoxic/lipotoxic conditions [25].

Chronic inflammation is a key factor of tissue damage in T2D. Pancreatic cells exposed to acute inflammation in the experiment display an adaptive response associated with sustained transcriptional and epigenetic reprogramming that leads to the activation of different gene expression programs [26]. However, a high level of inflammation within the pancreas greatly influences the outcome of pTF-mediated acinar cell reprogramming [27].

Macrophages are the key mediators of both tissue injury and repair during inflammation, including in T2D [28,29]. Persistent low-grade inflammation in adipose tissue and pancreas activates macrophages to produce pro-inflammatory products, which leads to dysfunction and destruction of islet β-cells and also inhibits the possible transdifferentiation of acinar and ductal cells in β-like insulin-producing cells. 5-amino-2.3-dihydrophthalazine-1.4-dione salts (APH) were described as immunomodulatory substances exhibiting anti-inflammatory properties [30,31,32]. In vitro, it was demonstrated that APH may contribute to a change in the macrophage polarization and profile of cytokines secreted by macrophages [33]. We revealed earlier that APH restores the β-cell mass and ameliorates hyperglycemia in alloxan diabetes [34]. Studies have shown that 5-amino-2.3-dihydrophthalazine-1.4-dione salts (APH) may reduce pancreatic inflammation. This suggests that APH is a suitable candidate for stimulating the generation of extra-islet IPCs in T2D.

In this study, we aimed to analyze in vivo changes in characteristics of extra-islet IPCs in normal, pathologic (experimental T2D), and APH-induced low-inflammatory conditions. As a model of T2D, we used streptozotocin-nicotinamide (Stz-NA) diabetes, which corresponds to T2D: stable moderate hyperglycemia, insulin resistance, and a slight decrease in insulin.

## 2. Results

### 2.1. Confirmation of T2D Development

The development of T2D in rats was confirmed by a biochemical study, HOMA-IR, and oral glucose tolerance test (OGTT). T2D increased plasma glucose concentration, glycated hemoglobin level, and insulin resistance, estimated on the HOMA-IR index, and decreased plasma insulin level (Figure 1a–d). Diabetic rats demonstrated poor glucose tolerance, which was reflected in a significantly higher concentration of glucose in blood 120 min after an oral glucose load observed in T2D groups compared with the ND group (*p* < 0.05) (Figure 1e). Elevated numbers of white blood cells (WBC) in the T2D cf. intact group (*p* < 0.05) (Figure 1f) were testified in favor of the development of inflammatory reactions and can be explained by the recruitment of WBC to the pancreas and other organs due to an autoimmune attack on glycosylated proteins. Thus, a state typical for T2D, including the formation of persistent moderate hyperglycemia, insulin resistance, and the development of a complex of inflammatory reactions, was simulated. Plasma level of corticosterone did not significantly changed (135.75 ± 3.12 pg/mL in ND group, 151.75 ± 7.03 pg/mL in T2D30, and 135.60 ± 6.04 pg/mL in T2D60).

### 2.2. Extra-Islet IPCs in the Pancreas of Non-Diabetic Rats

IPCs were found in the pancreas outside the islets of Langerhans (Figure 2e–i). Approximately 80% of these cells are localized in acini, the rest—in ducts (Figure 2a). Most extra-islet IPCs are gathered in small clusters (*n* ≤ 5) (Table 1). The average size of solitary acinar IPCs in non-diabetic (ND) rats is almost twice as much as ductal IPCs (*p* < 0.05) (Figure 2b, Table 1). IPCs gathered in small groups in both acini and ducts have similar size measures (Figure 2c, Table 1). The optical density of cytoplasm of IPCs in ND rats was the same regardless of the localization of IPC (in islets, acinar, or duct) for both solitary IPCs and IPCs in clusters (Figure 2d, Table 1).

### 2.3. The Effect of Type 2 Diabetes on the Pancreatic Extra-Islet IPCs

As expected, the total mass of IPCs in the pancreas of rats with T2D was lower than in ND animals (Table 1). However, extra-islet IPCs mass decreased in T2D less than islet β-cell mass: compared to ND rats, the quantity of β-cells decreased by 2,5-fold even by the 30th day of diabetes (*p* < 0.05), but the mass of extra-islet IPCs in both acini and ducts declined gradually, and differences from ND rats appeared only by 60th day of diabetes (*p* < 0.05) (Figure 1a, Table 1). The decline in the mass of extra-islet IPCs by 60th days of diabetes was due to a decrease in the number of clustering IPCs. The number of solitary extra-islet IPCs, compared to ND rats, changed neither in acini nor in ducts (Table 1). Moreover, by the 30th day of diabetes, the number of acinar solitary IPCs was increased cf. ND rats.

T2D did not change observed in ND rats. The ratio between acinar and ductal IPCs (Table 1) and predominance of the average size of solitary IPCs located in acini over the average size of solitary IPCs located in ducts (Figure 2b, Table 1). Moreover, by the 30th day of diabetes, the area of acinar solitary IPCs was increased cf. ND rats (*p* < 0.05).

T2D decreased the optical density of cytoplasm only in acinar IPCs located in clusters (*p* < 0.05) (Table 1).

### 2.4. Pro- and Anti-Inflammatory Cytokines in Blood and Pancreas

As expected, T2D increased the TNF-α (*p* < 0.05) and IFN-γ concentrations (*p* < 0.05) in blood and decreased the TGF-β1 concentration in both blood (*p* < 0.05) and pancreas (*p* < 0.05) (Figure 3b1,b2).

To reduce inflammation in T2D, we used APH. APH decreased IFN-γ concentration (*p* < 0.05) (Figure 3b1) and number of WBC (*p* < 0.05) (Figure 1f) in blood and TNF-α concentration in pancreatic tissue (*p* < 0.05) and increased TGF-β1 concentration in both blood (*p* < 0.05) and pancreas (*p* < 0.05) (Figure 3b1,b2).

### 2.5. Extra-Islet IPCs in the Pancreas of Diabetic APH-Treated Rats

APH increased the mass and optical density of cytoplasm of extra-islet IPCs, in both solitary and in groups in both acini and ducts cf. ND and diabetic rats (Figure 2d, Table 1). APH did not change the ratio between acinar and ductal IPCs (Table 1). The area of solitary ductal IPCs increased, and the area of ductal IPCs in clusters decreased in APH-treated diabetic rats compared to diabetic rats (Figure 2b,c, Table 1).

### 2.6. Characteristics of Glucose Metabolism in Diabetic APH-Treated Rats

APH-treated diabetic rats demonstrated reduced plasma glucose concentration (7.18 ± 0.19 mmol/L cf. 12.44 ± 0.26 mmol/L in diabetic rats) (Figure 1a). The decline of glycated hemoglobin level testified to the stability of the decrease in glucose level (Figure 1b). APH-treated diabetic rats had an increased level of blood insulin and reduced HOMA-IR parameter cf. rats with T2D (Figure 1c,d). The plasma level of corticosterone did not change (143.98 ± 5.00 pg/mL in APH-treated T2D rats cf. 135.60 ± 6.04 pg/mL in T2D60).

### 2.7. Expression of Pdx1 in Exocrine Pancreas

Expressions of Pdx1 were detected at the protein level by immunofluorescence analysis (Figure 3c4–c7). As expected, in ND rats, the number of Pdx1 immunoreactive (Pdx1^+^) cells in the islet of Lanhergance was 7–13-fold more cf. exocrine pancreas (Figure 3c1–c3). There were no differences in Pdx1 expression between acinar and ductal cells.

In T2D, the number of Pdx1^+^ cells was reduced in both the islets and exocrine pancreas. Expression of Pdx1 with a significant difference cf. ND rats in acini on the 60th day of diabetes and in duct epithelium—on the 30th day (Figure 3c1–c3).

Expression of Pdx1 in acini and ducts in diabetic APH-treated rats: Injections of APH resulted in the increased number of Pdx1-positive cells in acini and ducts cf. T2D 60 days in 2.5–3 (*p* < 0.05)- and 1.5–2 (*p* < 0.05)-fold, respectively. The quantity of Pdx1-positive cells in islets increased less than 1.5-fold (*p* < 0.05) (Figure 3c1–c3).

Thus, APH affected the number of Pdx1-positive cells in the non-endocrine pancreas, with a greater increase in acini (*p* < 0.05).

### 2.8. Quantity of F4/80^+^ Cells

The immunohistochemical analysis of monocyte-/macrophage-derived cells (F4/80^+^ cells) showed that macrophage infiltration of pancreatic acini increased in T2D (*p* < 0.05) and decreased in APH-treated diabetic rats (*p* < 0.05) (Figure 3a1–a3).

## 3. Discussion

In this study, we investigate the response of extra-islet IPCs to the development of experimental T2D and to i.m. administration of sodium 5-amino-2.3-dihydrophthalazine-1.4-dione (APH), which modulates the functional activity of macrophages, shifting them to reparative anti-inflammatory phenotype. We used Stz-NA diabetes as a model of T2D because it corresponds to the majority of main characteristics of T2D. In this study, for the first time, we compared the characteristics of the IPCs, localized in the epithelium of acini and ducts in non-diabetic, diabetic, and APH-treated diabetic rats. We showed that experimental diabetes crushed the mass of extra-islet IPCs less than islet IPSc (β-cells). At the same time, acinar and ductal extra-islet IPCs react differently to the development of experimental T2D and APH treatment. Table 1 summarizes the results obtained. The main findings are: 1. Pool of extra-islet IPCs in rats is not homogeneous. 2. Ductal IPCs demonstrate less variability cf. acinar IPCs in response to both development of T2D and APH treatment. 3. APH contributes to an increase in the extra-islet IPCs’ mass and optical density of cytoplasm in them.

### 3.1. Extra-Islet IPS in the Healthy Pancreas

Our study demonstrates that in the healthy pancreas IPCs, besides islets, are also localized in the epithelium of acini and ducts, consistently with [6,8,10,15]. According to our observation, the percentage of extra-islet IPCs from all pancreatic IPCs in rats is between 2.5% and 3%. We showed that the size of solitary extra-islet IPCs differs in acini and ducts; meanwhile, IPCs in clusters (n ≤ 5) even in different localizations have the same size. The optical density of cytoplasm in acinar and ductal IPCs is the same.

### 3.2. Changes in Extra-Islet IPCs, Promoted by Stz-NA Diabetes

Treatment of the animals with Stz decreased the mass of both islet and extra-islet IPCs, but for islet IPCs (β-cells), it is much more pronounced than for extra-islet IPCs. The difference in the rate of decreasing the IPCs’ quantity in islets and outside them, observed during the development of experimental diabetes, led to the fact that the proportion of extra-islet IPCs in the total number of IPCs increased, and by the 60th day of diabetes was about 10% (Table 1). We suppose that the increase in the number of solitary acinar IPCs observed on the 30th day of diabetes (Table 1) could be interpreted as a compensatory response from the pancreatic tissue to the development of hyperglycemia.

Thus, in healthy pancreas IPCs found outside islets of Langerhans in the epithelium of acini and ducts, solitary and in small groups, in diabetes, the number of extra-islet IPCs decreases; heterogeneity of the response of different subpopulations of extra-islet IPCs manifests itself through changes in size and functional activity of cells.

### 3.3. The Effect of APH on Extra-Islet IPCs

In APH-treated diabetic rats, an increase in the quantity of extra-islet IPCs and optical density of cytoplasm in them is observed.

We speculate that the modulatory effect of APH on extra-islet IPCs may be mediated by decreasing macrophage infiltration and inflammation in an Stz-damaged pancreas, which possibly allows expression of Pdx1, required for extra-islet IPCs growth-promoting, thus increasing the mass and functional activity of extra-islet IPCs. The rise of insulin blood level and weakening of hyperglycemia, observed in APH-treated diabetic rats, are in suitable agreement with an increase in the mass and functional activity of extra-islet IPCs and, in part, may be linked.

We suppose that inflammation in the pancreas caused by diabetic agent Stz first triggers processes aimed at restoring the functional integrity of the pancreatic tissue and then growing as diabetes develops, contributing to the dysfunction of IPCs. Probably, hyperglycemia can affect the generation of new IPCs: in studies in vitro, glucose stimulates the generation of insulin-producing cells from stem cells (reviewed in the work of [35]. At the early stage of diabetes, the expression of Pdx1 in acinar cells is still quite high, which allows to transdifferentiate them in IPCs. Probably, moderate inflammation in the pancreas at the beginning of diabetes development could contribute to the processes of transdifferentiation of non-β-cells to β-cells as a result of epigenetic reprogramming of non-endocrine pancreatic cells. However, as diabetes develops, macrophages infiltrate the exocrine pancreas and secrete pro-inflammatory cytokines, which reduces the ability of acinar cells to transdifferentiate into IPCs. Such a conclusion is consistent with previous studies that have shown the importance of balance in inflammatory levels and reprogramming of pancreatic transcriptional factors for the outcome of transdifferentiation of acinar cells into new β-like cells [27].

Since it is known that the presence of pro-inflammatory-activated macrophages in the pancreas blocks pTF-mediated reprogramming of acinar cells into β-like cells and may redirect the outcome of transdifferentiation into acinar-to-ductal metaplasia [27], we used APH, which decreases the infiltration of pancreatic tissue by macrophages and shifts the macrophage production to anti-inflammatory. Pancreatic epithelial cells react to inflammation by transcriptional and epigenetic reprogramming and activation of multiple gene expression programs [26,36]. Possibly, a change of inflammatory condition in the pancreas could trigger Pdx1 expression, with no outcome until the level of inflammation, mediated by macrophage production, remained sufficiently high. After reducing pancreatic macrophage infiltration and switching its production by APH, early activated pancreatocytes begin to show a greater Pdx1 expression, which results in increased transdifferentiation of non-endocrine cells into insulin-producing β-like cells. Anti-inflammatory macrophages dominate during pancreas repair and regeneration and, as a result of shifting of macrophage production by APH to anti-inflammatory, could facilitate the generation of new IPCs from progenitor cells located in pancreatic acini and ducts.

We suggest that an increasing concentration of TGF-β1 in the pancreatic tissue, observed in APH-treated diabetic rats, also contributes to increased generation of IPCs in the acinar and ductal epithelium, which is consistent with the work of [37], which demonstrated that TGF-β signaling affects not only β-cell development, proliferation, and function, but also control β-cell proliferation after β-cell loss. More, it suggests that inflammation-induced β-cell proliferation is TGF-β receptor signaling dependent [38].

The question remains open, whether different cell types have the same reprogramming potential. We suppose that currently, we do not have enough data to answer this question. However, our study shows that the number, size, and functional activity of acinar and ductal IPCs change differently both in rats with diabetes and APH-treated diabetic rats.

We are aware that we should not explain increased content of blood insulin and amelioration of hyperglycemia in APH-treated diabetic rats only by the generation of new extra-islet IPCs. Of course, insulin production could also increase due to islet β-cells stimulated by APH [34]. However, one cannot ignore the fact that the percentage of extra-islet IPCs from all pancreatic IPCs in APG-treated diabetic animals is greater cf. the control group. Moreover, extra-islet IPCs show greater stability in experimental diabetes cf. islet β-cells.

## 4. Materials and Methods

Experiments were performed on 40 12-week-old male Wistar rats weighing 310.0 ± 18.2 g for breeding in the animal facility of the Institute of Immunology and Physiology UB RAS (Yekaterinburg, Russia). Rats were housed 5 per standard polypropylene cage and were kept under equal conditions (12 h light (from 9:00 a.m. to 9 p.m.): 12 h dark cycle, temperature 22 ± 2 °C above zero). Animals were fed according to the customary schedule by soy protein-free extruded rodent diets (2020X Teklad, Envigo, Huntingdon, UK) and were provided filtered tap water *ad libitum*.

### 4.1. Induction of Experimental Diabetes

Experimental diabetes was modeled by a single i.p. injection of 65 mg/kg b.w. streptozotocin (Stz) (Sigma-Aldrich, St Louis, MO, USA, Cat. No. S0130), 15 min after the injection of 110 mg/kg b.w. of nicotinamide (NA) (Sigma-Aldrich, St Louis, MO, USA, Cat. No. N3376-100G) [39]. The dry powder of Stz was dissolved in 0.1 M citrate buffer (pH 4.5), and NA was dissolved in distilled water. All animals received fresh solutions of drugs after 16 h of fasting.

### 4.2. Experimental Protocol

A total of 30 Stz-NA-induced diabetic rats were randomly divided into three groups (n = 10 in each group): the 1st and the 2nd—duration of diabetes was 30 and 60 days, correspondingly (such duration of experimental diabetes is sufficient time for the development of diabetes complications [40]). The 3rd group—diabetic rats, treated with 5-amino-2.3-dihydrophthalazine-1.4-dione sodium salt at a dose of 2 mg/kg b.w. between 30 and 60 days, according to the experimental schedule (20 i.m. injections in total). Sodium 5-amino-2.3-dihydrophthalazine-1.4-dione (APH), (PubChemCID 9794222, InChI Key: JKEBMURXLKGPLR-UHFFFAOYSA-N) regulates the biosynthetic processes in cells, partially in macrophages, and suppresses production of anti-inflammatory cytokines in vivo and in vitro [30,31,32,33]. At the end of the experiment, rats were euthanized by 15 mg/kg b.w. Zoletil-100 (Virbac, Carros, France).

### 4.3. Fasting Blood Glucose (FBG) Level, Glycosylated Hemoglobin (HbA1c), and Oral Glucose Tolerance Test (OGTT)

Blood samples were collected before euthanasia after a 16 h fasting period from the tail vein and centrifuged at 500× *g* for 10 min at 4 °C. FBG level was determined by the glucose oxidase method using a ready diagnostic kit (Vektor-Best, NovosibirskRussia). The level of glycosylated hemoglobin (HbA1c) was measured using a ready kit for affinity chromatography (GLICOHEMOTEST, ELTA, Moscow, Russia). To evaluate glycemic control in animals with T2D, oral glucose tolerance tests (OGTT) were performed in the 4th week of the experiment. The assessment was based on the FBG level at 0 min and postprandial glucose (PG) level measured after oral administration of glucose (1 g/kg) at 30, 60, and 120 min. Administration of glucose was conducted using oral dosing needles (VetTech Solutions Ltd., Congleton, U.K.). The area under the curve (AUC) was calculated according to Sakaguchi et al. (2016) using the trapezoidal approximation of PG levels [41].

### 4.4. Insulin Level and HOMA-Estimated Insulin Resistance

Plasma insulin level was measured using a Rat Insulin ELISA kit (Thermo Fisher Scientific, Waltham, MA, USA). Insulin resistance was estimated by HOMA according to the method described by Matthews et al. [42]. It was calculated by the formula: HOMA-IR = fasting insulin (μU/mL) × fasting glucose (mmol/L)/22.5.

### 4.5. Preparation of Tissue Samples

A median laparotomy was performed under general anesthesia. The samples of pancreatic tissue were excised and fixed in 10% neutral formalin at room temperature overnight, processed for customary histological evaluation, and embedded into paraffin. Tissue sections of 3–4 mm thick were made using a microtome.

### 4.6. Immunohistochemical Evaluation of Pancreatic Tissues

Pancreatic paraffin sections were studied using immunohistochemistry (IHC) and fluorescent IHC. Insulin^+^ cells and macrophages (F4/80^+^ cells) were detected by IHC, Pdx1^+^—using fluorescent IHC. Tissues were labeled with primary antibodies (Table 2) overnight at 4 °C above zero, followed by incubation with secondary antibodies for 1 h. The immunohistochemical procedure was performed using the avidin-biotin peroxidase complex method. Staining was performed according to the manufacturers’ standard protocols. To check the protocol and exclude nonspecific staining, negative and positive controls were set. Sections of the pancreas of intact rats were a positive control [43,44,45].

### 4.7. Morphometric Analysis

Using pancreatic sections stained with IHC antibodies to proinsulin and insulin, insulin^+^ cells (IPCs) were detected. The number of islet IPCs (β-cells) and extra-islet IPCs in the epithelium of acini and ducts, located solitary or in small clusters of less than 5 cells [6], was estimated per unit area (N/mm^2^). Areas of extra-islet IPCs were determined. Based on the detection of optical density of cytoplasm, the content of insulin and functional activity of IPCs were assessed.

The number of macrophages in the pancreatic tissue was estimated on sections stained with an F4/80 marker. Assessment of Pdx1 expression in the pancreas was made based on the quantity of Pdx1^+^ cells.

### 4.8. Microscopic Examination

Tissue slides examination was conducted using the light microscope (Leica DM 2500), and image analysis was performed using Video TesT-Morphology 5.0 software (VideoTesT, St. Petersburg, Russia). Visualization of Pdx1^+^ cells was made on a confocal laser scanning microscope Carl ZEISS LSM 710 (Carl Zeiss, Germany).

### 4.9. Inflammatory Characteristics and Cytokines Content

The quantity of leukocytes (WBC) in heparinized blood samples was estimated using an automated hematology analyzer Celly 70 (Biocode Hycel, France). The content of corticosterone was determined in plasma, and levels of pro-inflammatory cytokines TNF-α, IFN-γ, and anti-inflammatory TGF-β1 were determined in plasma and pancreas by ELISA using LAZURITE Automated ELISA System (Dynex, USA). A list of the used kits is in Table 2.

### 4.10. Statistical Data Analysis

Quantitative data were presented as mean ± SEM. Statistical analysis was performed using Microsoft Office Excel 2007 and Origin Pro 9.0 software. Differences among groups were analyzed by the Mann–Whitney U test, and a *p*-value less than 0.05 was considered statistically significant.

## 5. Conclusions

We concluded that some pancreatic acinar and ductal cells could act as a functional reserve of insulin production in the case of islet β-cells dysfunction in experimental diabetes. Additionally, we observed that the mass of extra-islet IPCs may be increased in diabetic rats by decreasing the degree of macrophage infiltration and inflammatory level in pancreatic exocrine tissue due to APH treatment. More in-depth research is needed in order to further elucidate the molecular mechanisms of the modulation action of APH on macrophages in a T2D rat model. These are very important for the development of new therapeutic agents for patients with T2D.

## 6. Limitations

The main limitation of the study was there we did not investigate the action of insulin secretion by extra-islet IPCs: is it endocrine, paracrine, or autocrine? Therefore, the role of extra-islet IPCs in pancreas regeneration and restoration of its insulin-production function is still unclear. Future studies will examine this issue to establish the role of pancreatic extra-islet IPC and, widely, IPCs in non-pancreatic tissues. Second, we did not evaluate the impact of stem cells in extra-islet IPCs generation.

## Figures and Tables

**Figure 1 ijms-23-04286-f001:**
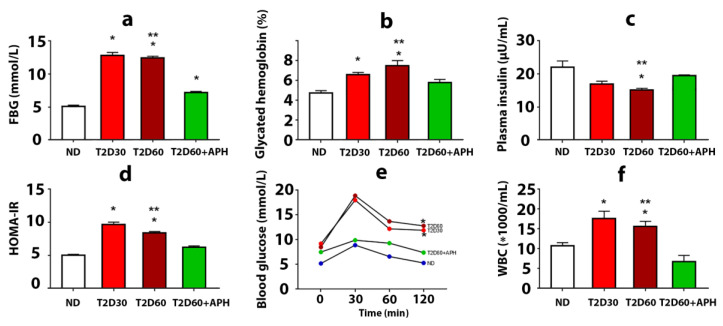
The effects of 5-amino-2.3-dihydrophthalazine-1.4-dione salts (APH) treatment on rats with experimental type 2 diabetes (T2D). The fasting blood glucose level (**a**), concentration of glycated hemoglobin (**b**), mean of HOMA-IR index (**d**), and quantity of white blood cells (WBC) (**f**) showed a decrease in APH-treated T2D rats compared to rats with T2D60; conversely, plasma insulin level (**c**) increased. Oral glucose tolerance test (OGTT) (**e**) showed no difference between blood glucose concentration before and 120 min after 1 g oral glucose load in APH-treated diabetic animals. ND—non-diabetic rats, T2D30—rats with T2D lasts 30 days, T2D60—rats with T2D lasts 60 days, T2D60 + APH—rats with T2D lasts 60 days, treated with APH; *n* = 10 in each group. Data are mean ± SEM. (**a**–**e**): *—*p* < 0.05 in comparison with ND; **—*p* < 0.05 in comparison with T2D60+APH; (**f**): *—*p* < 0.05 in comparison with baseline (t = 0 min).

**Figure 2 ijms-23-04286-f002:**
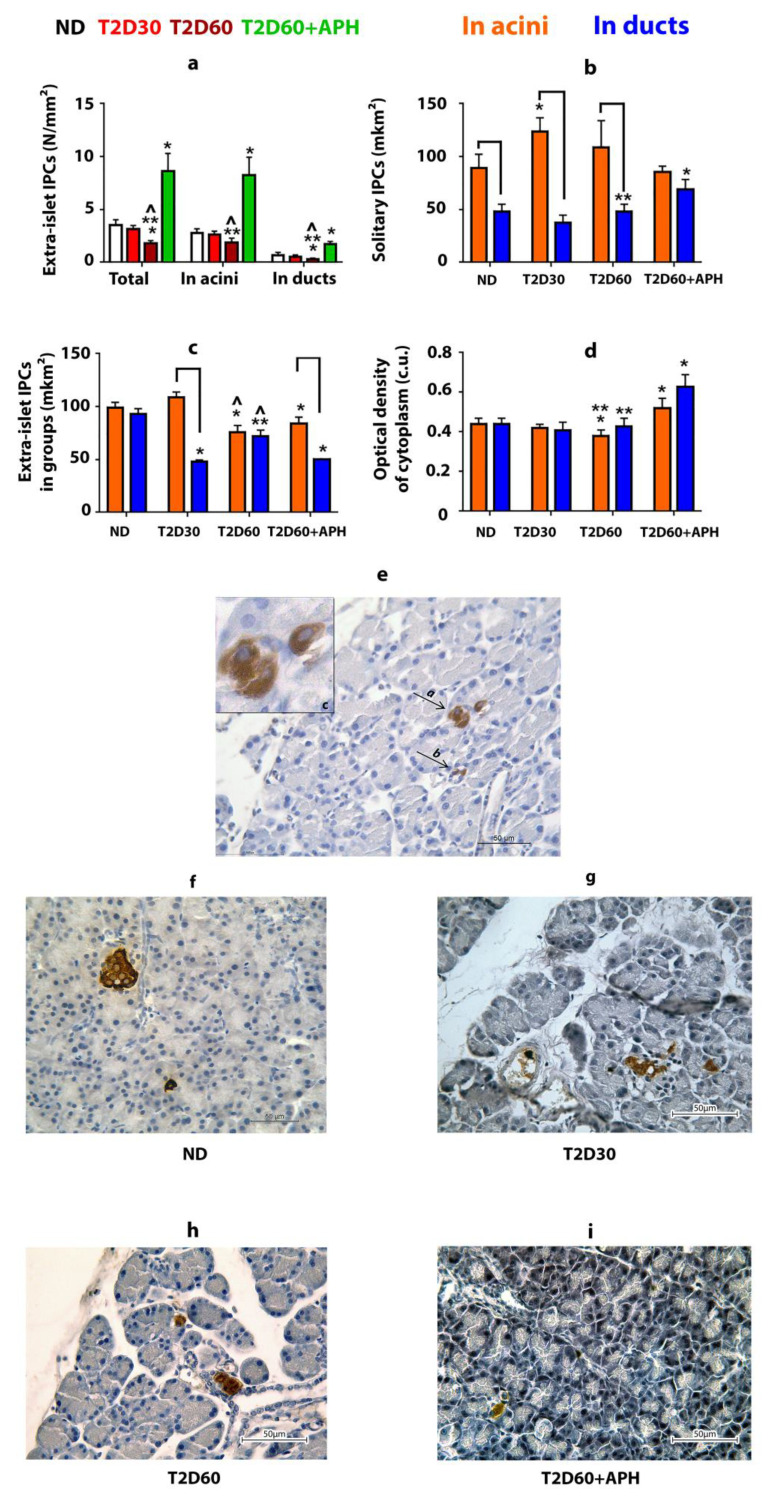
Extra-islet insulin-producing cells (IPCs) in healthy, diabetic, and APH-treated diabetic rats. APH promotes an increase in the number of extra-islet IPCs (**a**) and the optical density of insulin in them (in conventional units—c.u.) (**d**) in both acini and ducts. Areas of acinar IPCs are bigger than ductal IPCs, as in the case of solitary located IPCs (**b**), as in small groups of IPCs (**c**,**e**). (**e**–**i**)—immunolocalization of insulin in the pancreas of ND (**f**), T2D30 (**g**), T2D60 (**h**), and APH-treated T2D60 (**i**) rats (brown), light microscopy. **e**(a)—IPCs in acini, **e**(b)—IPCs in ducts, **e**(c)—in acini, enlarged. ND—non-diabetic rats, T2D30—rats with T2D lasts 30 days, T2D60—rats with T2D lasts 60 days, T2D60 + APH—rats with T2D lasts 60 days, treated with APH; *n* = 10 in each group. Data are mean ± SEM. *—*p* < 0.05 in comparison with ND; **—*p* < 0.05 in comparison with T2D60 + APH; ^—*p* < 0.05 in comparison with T2D30; **⎴** —*p* < 0.05 between acinar and ductal IPCs.

**Figure 3 ijms-23-04286-f003:**
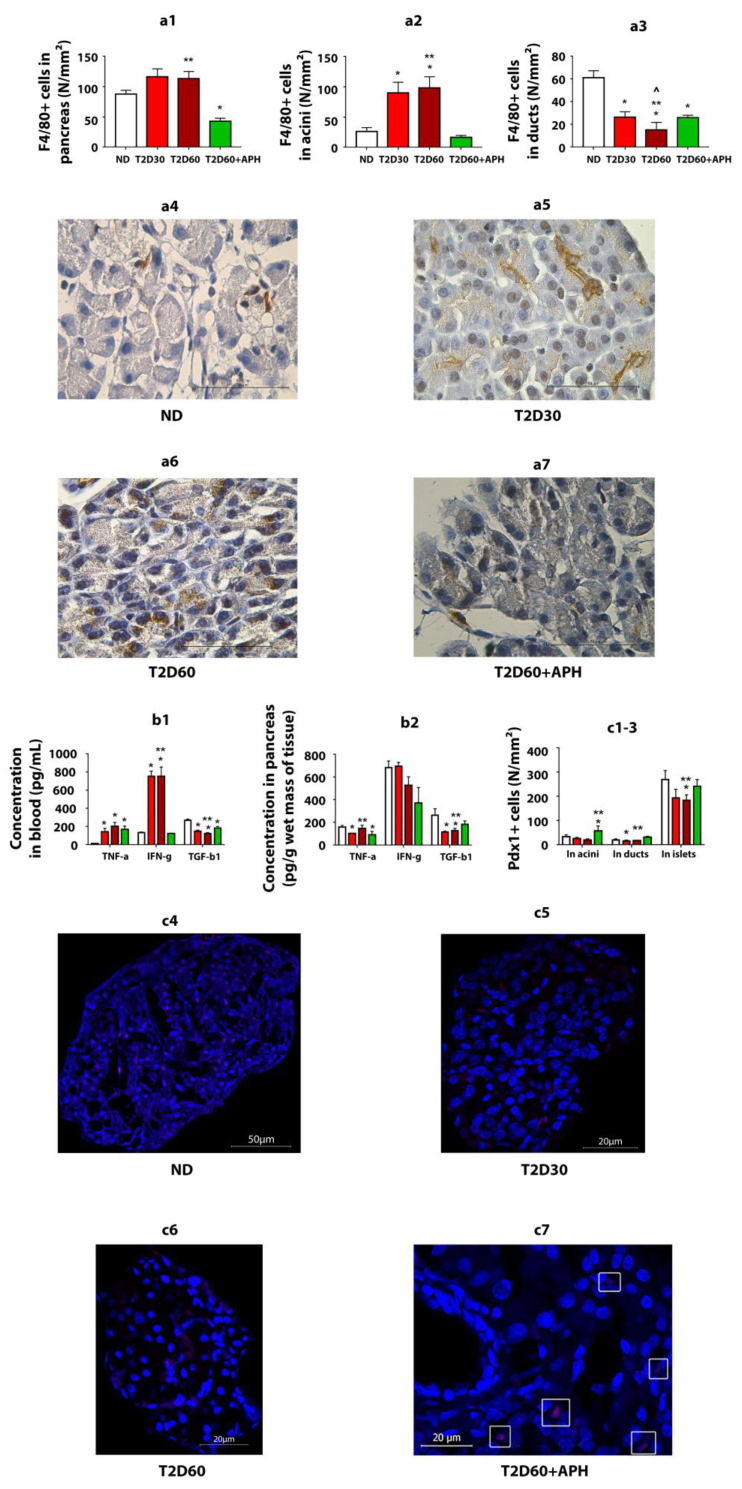
APH reduces the number of macrophages (F4/80+ cells) in the pancreas of T2D rats, changes cytokine concentration in blood and pancreas of T2D rats, and stimulates the expression of Pdx1 in the exocrine pancreas of T2D rats. (**a1**)—total amount of F4/80-positive (^+^) cells in exocrine pancreas, in particular: (**a2**)—in acini, (**a3**)—in ducts; (**a4**–**a7**)—immunohistochemistry for macrophage marker F4/80 in rat pancreas, light microscopy. Concentration of cytokines: (**b1**)—in blood, (**b2**)—in pancreas. Quantity of Pdx1-positive (^+^) cells: (**c1**)—in acini, (**c2**)—in ducts; (**c3**)—in islets. *n* = 10 in each group. (**c4**–**c7**)—immunofluorescence staining of the rat pancreas for Pdx1, confocal laser microscopy; in the pancreas of ND (**c4**), T2D30 (**c5**), and T2D60 (**c6**) rats, Pdx1^+^ cells are mainly localized in islets, in APH-treated T2D rats the presence of Pdx1^+^ cells in exocrine acinar (**c7**, surrounded by a white frame) increases. Data are mean ± SEM. *—*p* < 0.05 in comparison with ND; **—*p* < 0.05 in comparison with T2D60 + APH; ^—*p* < 0.05 in comparison with T2D30.

**Table 1 ijms-23-04286-t001:** Mass, area, and optical density of cytoplasm of IPCs (mean ± SEM).

	ND	T2D30	T2D60	T2D60 + APH
IPCs/mm ^2^				
Total IPCs mass	169.82 ± 43.23	63.45 ± 12.97 ^1^	51.52 ± 15.27 ^1^	120.52 ± 27.03 ^3^
Islet IPCs (β-cells)	166.32 ± 42.76	63.3 ± 15.88 ^1^	49.72 ± 12.35 ^1^	111.87 ± 20.71 ^3^
Extra-islet IPCs	3.5 ± 0.54	3.15 ± 0.34	1.81 ± 0.24 ^1,2^	8.65 ± 1.66 ^1,3^
Among them (*IPCs in acini from all extra-islet IPCs, %*):				
In acini	2.8 ± 0.37 *(81.05 ± 5.02)*	2.61 ± 0.33 *(82.98 ± 5.39)*	1.86 ± 0.43 ^2^ *(78.28 ± 7.86)*	8.23 ± 1.71 ^1,3^ (76.51 ± 6.89)
In ducts	0.71 ± 0.24	0.54 ± 0.16	0.28 ± 0.12 ^1,2^	1.74 ± 0.24 ^1,3^
Including:				
solitary IPCs:				
In acini	0.28 ± 0.09	0.78 ± 0.14 ^1^	0.28 ± 0.09 ^2^	1.94 ± 0.36 ^1,3^
In ducts	0.12 ± 0.05	0.11 ± 0.04	0.17 ± 0.07	0.52 ± 0.06 ^1,3^
IPCs in groups:				
In acini	2.51 ± 0.34	1.83 ± 0.27	1.58 ± 0.42 ^1^	6.29 ± 2.05 ^1,3^
In ducts	0.59 ± 0.19	0.43 ± 0.15	0.09 ± 0.06 ^1^	1.22 ± 0.28 ^1,3^
In acini	81.05 ± 5.02 *	82.98 ± 5.39 *	78.28 ± 7.86 *	76.51 ± 6.89 *
In ducts	18.95 ± 5.02	17.02 ± 5.39	21.72 ± 7.86	23.49 ± 6.89
Area of extra-islet IPCs, mkm^2^:				
Solitary IPCs:				
In acini	89.23 ± 12.97 *	123.49 ± 13.25 ^1,^*	109.03 ± 24.98 *	85.8 ± 5.1 ^2^
In ducts	47.65 ± 7.45	37.2 ± 7.45	47.8 ± 7.31	68.7 ± 9.7 ^1,2,3^
IPCs in groups:				
In acini	98.69 ± 5.51	108.88 ± 4.98	76.03 ± 6.18 ^1,2^	83.88 ± 6.24 ^1^
In ducts	92.97 ± 5.26	47.88 ± 1.68 ^1,^*	72.34 ± 5.54 ^2^	49.98 ± 0.04 ^1,3,^*
Optical density of IPCs’ cytoplasm, conventional units:				
Solitary IPCs:				
In acini	0.43 ± 0.04	0.41 ± 0.03	0.36 ± 0.02	0.5 ± 0.05 ^3^
In ducts	0.44 ± 0.04	0.39 ± 0.02	0.35 ± 0.05	0.69 ± 0.07 ^1,2,3^
IPCs in groups:				
In acini	0.45 ± 0.01	0.43 ± 0.02	0.36 ± 0.02 ^1,2^	0.55 ± 0.03 ^1,2,3^
In ducts	0.44 ± 0.02	0.44 ± 0.06	0.44 ± 0.03	0.58 ± 0.02 ^1,2,3^
Islet IPCs (β-cells)	0.41 ± 0.02	0.38 ± 0.03	0.35 ± 0.02 ^1^	0.53 ± 0.08 ^1,2,3^

^1^—*p* < 0.05 in comparison with ND; ^2^—*p* < 0.05 in comparison with T2D30; ^3^—*p* < 0.05 in comparison with T2D60; *—*p* < 0.05 in comparison with ductal IPCs within the same group.

**Table 2 ijms-23-04286-t002:** List of antibodies for immunohistochemical and immunofluorescence studies and ELISA kits used in the study.

Detected Antigen	Primary Antibodies: Reference, Supplier, Dilution	Secondary Antibodies: Reference, Supplier, Dilution
Proinsulin and insulin	Anti-Insulin/Proinsulin: clone INS04+INS05, MA5-12042, Invitrogen, Carlsbad, CA, USA, 1:200	Biotin Goat anti-Mouse Ig (Multiple Absorption), BD Pharmingen, San Diego, USA, 1:500
Pdx1	Anti-PDX1: ab 227586, Abcam, Branford, CT, USA, 1:200	Goat anti-Rabbit IgG (H+L) + Texas Red, Thermo Fisher Scientific, Waltham, MA, USA, 1:100
F4/80	Anti-F4/80 Polyclonal Antibody, PA5-21399, Thermo Fisher Scientific, Waltham, MA, USA, 1:200	Biotin Goat Anti-Rabbit IgG, Thermo Fisher, Scientific, Waltham, MA, USA, 1:50
**Detectable protein**	**ELISA Kit, supplier**	
Insulin	Rat Insulin ELISA Kit, Invitrogen-Thermo Fisher Scientific, Waltham, MA, USA	
Corticosterone	Corticosterone ELISA Kit, Abcam, Cambridge, Great Britain	
TNF-α	TNF alpha Rat ELISA Kit, Invitrogen-Thermo Fisher Scientific, Waltham, MA, USA	
IFN-γ	IFN gamma Rat ELISA Kit, Invitrogen-Thermo Fisher Scientific, Waltham, MA, USA	
TGF-β1	TGF beta-1 Rat ELISA Kit, Invitrogen-Thermo Fisher Scientific, Waltham, MA, USA	

## Data Availability

Not applicable.

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
