# Peer review of "Accelerated Generation of Extra-Islet Insulin-Producing Cells in Diabetic Rats, Treated with Sodium Phthalhydrazide"

_ijms, 2022, doi:10.3390/ijms23084286_

Round 1

Reviewer 1 Report

This manuscript investigates the role of APH in generation of IPCs in type 2 diabetic rats. This manuscript is well written and poses an interesting insight to the potential role of APH in type 2 diabetes. However, this study needs greater scientific rigour and methodology:

  • Please label the y axis on all the graphs instead of just putting the units. It is very distracting to have to keep referring to the figure legends to know what the graphs are showing.
  • Please increase the label size on each graph and include the n numbers for each group in the figure legends.
  • How did the authors quantify white blood cells in figure 1? What staining did the authors use to elucidate these cells?
  • Please elaborate more and show how the APH shifts the pro-inflammatory macrophages to more reparative anti-inflammatory macrophages as suggested in the discussion.
  • The authors should include the F4/80 macrophage staining for every group in the acini, ducts and whole pancreas.
  • Cytokine expressions done in the blood. Please specify whether the ELISAs were done from the whole blood or whether from the serum or plasma as each will yield different results.
  • The authors showed some of the main pro-inflammatory cytokine profiles in their study. However, more cytokines are encouraged to be included as well such as IL-6 and IL1 beta as they are main pro-inflammatory cytokines as well. Anti-inflammatory cytokine such as IL10 is encouraged to be included in order to get the full picture of how APH regulates cytokine expressions both in diabetic blood and pancreas.
  • The authors should quantify Pdx expression through more quantitative methods and experiments instead of through counting Pdx positive cells to provide a clearer evidence. Also is Pdx solely expressed by IPCs? In their IHCs, the authors should include a staining to show which cells are expressing Pdx. Please provide this for every group treatment as well.
  • Also do the authors include the staining for insulin/pro-insulin marker in the manuscript? Please clarify.
  • The level of TNF alpha in the pancreas of the ND group is as high as the T2D60 group. Please explain this since it suggests that the ND pancreas is in the same inflammatory state as the diabetic group. Similarly, with the IFN gamma between ND and T2D30 group.
  • It is rather confusing as to what the authors are suggesting with the increased levels of TGF beta in the blood and pancreas. Is TGF beta in this study acting as pro or anti inflammatory since the authors discusses that TGF beta is contributing to the proliferation and the loss of beta cells but also inducing inflammation. Please clarify on this.
  • The use of APH is also a bit confusing as the authors describe that APH suppresses the production of anti-inflammatory cytokines. Please elaborate more on how APH might be beneficial in T2D and in your animal model as this sounds contradictory to what this study is trying to achieve.
  • Observing from the graphs on the macrophage numbers, it appears that the number of macrophages in the acini is significantly higher than in the ducts. The authors should elaborate more why this might occur. Is there a chance that there is macrophage migration occurring in the pancreas?

Generally, this is a very interesting study, however, the authors should show more functional data in conjunction with counting positive cells and ELISAs.

Author Response

Dear colleagues,

We would like to thank you for your helpful comments. Please find below a point-by-point response to your comments and questions.

Comments of Reviewer â„– 1:

Point 1: Please label the y axis on all the graphs instead of just putting the units. It is very distracting to have to keep referring to the figure legends to know what the graphs are showing.

Response 1: We agree with the reviewer. We labeled the y axis on all graphs.

Point 2: Please increase the label size on each graph and include the n numbers for each group in the figure legends.

Response 2: We increased the label sizes on each graph and included the n number of animals in each group in the figure legends, as suggested by the reviewer.

Point 3: How did the authors quantify white blood cells in figure 1? What staining did the authors use to elucidate these cells?

Response 3: Quantity of leukocytes (WBC) in heparinized blood samples were estimated using automated hematology analyzer Celly 70. This information is included in 4.9 (lines 399-400).

Point 4: Please elaborate more and show how the APH shifts the pro-inflammatory macrophages to more reparative anti-inflammatory macrophages as suggested in the discussion.

Response 4: The changes in macrophage production towards anti-inflammatory cytokines during treatment with APH were demonstrated in [30-34] (lines 92-98), but the mechanisms of it are required further molecular and genetic studies. In the current study we used APH to illustrate how the shift of macrophage production may effect at the function activity of pancreatic cells by example of extra-islet insulin-producing cells.  

Point 5: The authors should include the F4/80 macrophage staining for every group in the acini, ducts and whole pancreas.

Response 5: We included the F4/80 macrophage marker staining for every groups in the Figure 3.

Point 6: Cytokine expressions done in the blood. Please specify whether the ELISAs were done from the whole blood or whether from the serum or plasma as each will yield different results.

Response 6: For ELISAs we used plasma, we specify it in the manuscript (4.9, line 402), according to reviewer suggestion.

Point 7: The authors showed some of the main pro-inflammatory cytokine profiles in their study. However, more cytokines are encouraged to be included as well such as IL-6 and IL1 beta as they are main pro-inflammatory cytokines as well. Anti-inflammatory cytokine such as IL10 is encouraged to be included in order to get the full picture of how APH regulates cytokine expressions both in diabetic blood and pancreas.

Response 7: According to Chen T. et al. (2015), changes in the level of cytokines IL6 and IL1β in DM2 occur in the same direction as TNFα. Shift of the macrophage phenotype to anti-inflammatory may be verified based on increase in the level of IL10 and TGFβ1 and decreases in the level of IL6, IL1β1, TNFα (Zhang X. et al., 2015; Shapouri-Moghaddam A. et al., 2018).  From the group of the proinflammatory cytokines TNFα was chosen as the most integrated indicator, associated not only with inflammation but also with insulin resistance in T2D. The choice of TGFβ1 was due to its involvement in the development and proliferation of β-cells [F. Sanvito et al., 1994; X. Xiao et al., 2014; N. Van Gassen et al., 2015]. However, we agree with the reviewer that the determination of IL6, IL1β and IL10 is of interest. As for IL6, a decrease in its content in the blood was shown in the APH-treated rats with alloxan diabetes [34]. In future work, we plan to determine a wider range of cytokines.

Chen T, Gao J, Xiang P, Chen YJi JXie P, Wu H, Xiao W, Wei Y, Wang SLan LJi H, Yan T. Protective effect of platycodin D on liver injury in alloxan-induced diabetic mice via regulation of Treg/Th17 balance//Int Immunopharmacol. 2015 Jun;26(2):338-48.

Zhang X, Zhou MGuo Y, Song Z, Liu B. 1,25-Dihydroxyvitamin D₃ Promotes High Glucose-Induced M1 Macrophage Switching to M2 via the VDR-PPARγ Signaling Pathway//Biomed Res Int. 2015;2015:157834. doi: 10.1155/2015/157834.

Shapouri-Moghaddam A, Mohammadian S, Vazini H, Taghadosi M, Esmaeili SA, Mardani F, Seifi B, Mohammadi A, Afshari JT, Sahebkar A. Macrophage plasticity, polarization, and function in health and disease. J Cell Physiol. 2018 Sep;233(9):6425-6440. doi: 10.1002/jcp.26429.

Concise Review: Macrophages: Versatile Gatekeepers During Pancreatic β-Cell Development, Injury, and Regeneration / N. Van Gassen, W. Staels, E. Van Overmeire, [et al.] // Stem cells translational medicine. – 2015. – Vol. 4 (6). – P. 555-563. https://doi.org/10.5966/sctm.2014-0272.

M2 macrophages promote beta-cell proliferation by up-regulation of SMAD7 / X. Xiao, I. Gaffar, P. Guo, [et al.] // Proceedings of the National Academy of Science USA. – 2014, Apr 1. – Vol. 111 (13). – P. E1211-20.

TGF-beta 1 influences the relative development of the exocrine and endocrine pancreas in vitro / F. Sanvito, P.L. Herrera, J. Huarte, [et al.] // Development. –1994. – Vol. 120. – P. 3451-3462.

Point 8: The authors should quantify Pdx expression through more quantitative methods and experiments instead of through counting Pdx positive cells to provide a clearer evidence. Also is Pdx solely expressed by IPCs? In their IHCs, the authors should include a staining to show which cells are expressing Pdx. Please provide this for every group treatment as well.

Response 8: IHC visualization of Pdx1 allows to evaluate the number of cells, expressing this marker. When comparing the optical density of cytoplasm in Pdx1+ cells to assess the quantitative content of this transcription factor, no significant differences were found between the experimental groups, so these data were not included in the manuscript.

We did not use double labeling for Pdx1 and insulin, because we considered Pdx1 expression as an indication of the readiness of cells to synthesize insulin. Furthermore, ectopic expression of Pdx1 alone can induce incomplete reprogramming of pancreatic acinar cells (Heller et al. 2001).

Heller RS, Stoffers DA, Bock T, Svenstrup K, Jensen J, Horn T, Miller CP, Habener JF, Madsen OD, Serup P 2001. Improved glucose tolerance and acinar dysmorphogenesis by targeted expression of transcription factor PDX-1 to the exocrine pancreas. Diabetes 50: 1553–1561

Point 9: Also do the authors include the staining for insulin/pro-insulin marker in the manuscript? Please clarify.

Response 9: We agree with reviewer. We add the staining of pancreatic slides for insulin/pro-insulin for all animal groups in the Figure 2. 

Point 10: The level of TNF alpha in the pancreas of the ND group is as high as the T2D60 group. Please explain this since it suggests that the ND pancreas is in the same inflammatory state as the diabetic group. Similarly, with the IFN gamma between ND and T2D30 group.

Response 10: We calculated the concentrations of cytokines in the pancreas per unit wet mass of the organ. This form of data presentation was chosen to minimize the possible error when recalculating the content of cytokines per 1 g of protein. We understand that this may have affected the results, namely pancreatic edema in T2DM may have resulted in lower values. To emphasize that reported data are not absolute values, we have added the relevant information about this methodological limitations in Figure 3.

Point 11: It is rather confusing as to what the authors are suggesting with the increased levels of TGF beta in the blood and pancreas. Is TGF beta in this study acting as pro or anti inflammatory since the authors discusses that TGF beta is contributing to the proliferation and the loss of beta cells but also inducing inflammation. Please clarify on this.

Response 11: We consider TGF-β1 as a growth factor and anti-inflammatory cytokine. TGF-β11 has inhibitory activity against T- and B-cell proliferation, inhibits the activity of NK cells, CD8+-lymphocytes, and thus helps to reduce the inflammatory response. TGFβ controls proliferation, cell differentiation in most cells. Despite the ability of TGFβ1 to enhance fibrogenesis and atherogenesis, this factor, as one of the growth factors, can be used to assess the change in the macrophage phenotype to anti-inflammatory.

In the lines 307-308 we mean that TGF-β1 is required for the proliferation of β cells, induced by inflammation, but which occurs only after the completion of the action of inflammatory factors.

Point 12: The use of APH is also a bit confusing as the authors describe that APH suppresses the production of anti-inflammatory cytokines. Please elaborate more on how APH might be beneficial in T2D and in your animal model as this sounds contradictory to what this study is trying to achieve.

Response 12: In [30-34] was shown that APH contributes to a change in the production of macrophages towards anti-inflammatory. Early we demonstrated [34] that APH decreases macrophage infiltration of alloxan-damaged pancreas and reprograms remaining macrophages to produce specific signals, possibly required for β-cell growth. In our work, we used APH to demonstrate how a decrease in pro-inflammatory macrophages in the pancreas and an increase in the production of anti-inflammatory growth factor TGF-β1 affects the number and some characteristics of insulin-producing cells in the acini and ducts of the pancreas.

Point 13: Observing from the graphs on the macrophage numbers, it appears that the number of macrophages in the acini is significantly higher than in the ducts. The authors should elaborate more why this might occur. Is there a chance that there is macrophage migration occurring in the pancreas?

Response 13: When islets were exposed to a hyperglycemia or in increased content of non-esterified fatty acids in T2D, increased islet-derived inflammatory factors were produced, released and attract macrophages to the islets (Ehses J.A. et al., 2007). However, inflammatory stimuli promote oxidative stress also in pancreatic acinar cells (Pan L. 2018), which should also attract macrophages to it.  Moreover, in T2D humans 33.3±13.6% of double positive for amylase and insulin cells in exocrine part of the pancreas were adjacent to macrophages and/or mast cells (Masini M. 2017). Thus, the increased exit of macrophages into the acini in T2DM is quite understandable, and among them can be both monocytes/macrophages recruited from the blood and resident pancreatic macrophages. At the same time, probably, in the ducts, in which inflammation is less than in islets or acini, the number of macrophages in T2DM is lower than in acini. With a decrease in pancreatic inflammation in APH-treated diabetic rats, the migration of macrophages into the acini also decreases, which automatically leads to an increase in their number in the ducts, since the resident macrophages of the ducts remain in them.

Increased number of islet associated macrophages in type 2 diabetes / JA Ehses, A. Perren, E. Eppler [et al. ] // Diabetes. - 2007. - Vol. 56. - P. 2356-2370

Pan L, Yu L, Wang L, He J, Sun J, Wang X, Wang H, Bai Z, Feng H , Pei H. Inflammatory stimuli promote oxidative stress in pancreatic acinar cells via Toll-like receptor 4/nuclear factor-κB pathway Int J Mol Med 2018 Dec;42(6):3582-3590 doi: 10.3892/ijmm.2018.3906 Epub 2018 Oct 1. PMID: 30272284

Masini M, Marselli L, Himpe E, Martino L, Bugliani M, Suleiman M, et al. (2017) Co-localization of acinar markers and insulin in pancreatic cells of subjects with type 2 diabetes. PLoS ONE 12(6): e0179398. https://doi.org/10.1371/journal.pone.0179398

Reviewer 2 Report

How do you justify a dose of STZ + NA, as a mouse model for type II diabetes? Did you feed the HFD to the rats?

I know some researchers have published articles before.

The mechanistic part of APH is completely lacking in the manuscript.

The authors should Include the plasma concentration of 2-methoxyestradiol and catechol-o-methyl-transferase levels in the treatment group.

What is the phosphorylation levels of AMPK in these extra-Islet Insulin Producing cells treated with APH?

Include one positive control group in your studies such as metformin. 

Author Response

Point 1: How do you justify a dose of STZ + NA, as a mouse model for type II diabetes? Did you feed the HFD to the rats?

Response 1: The streptozotocin-nicotinamide model we used reproduces T2DM without the development of obesity and is characterized by the development of stable moderate fasting hyperglycemia, a decrease in the number of β-cells and a decrease in insulin production, impaired glucose tolerance, polyphagia and polydipsia (Islam M.S. et al., 2012; Ghasemi A. et al.,2014). We did not use a fat diet, and the model and dose were verified based on plasma glucose and insulin levels, glycated hemoglobin levels, HOMA index, and glucose tolerance test (lines 107-119, Figure 1). A moderate deviation of these indicators from the norm allows us to attribute the condition obtained as a result of modeling to the late form of type 2 diabetes mellitus.

Ghasemi, A. Streptozotocin-nicotinamide-induced rat model of type 2 diabetes / A. Ghasemi, S. Khalifi, S. Jeddy // Acta physiologica Hungarica. – 2014. – Vol. 101(4). – P. 408-20. 10.1556/APhysiol.101.2014.4.2.

Islam, M.S.  Experimentally induced rodent models of type 2 diabetes / M.S. Islam, R.D. Wilson // MethodsMol Biol. – 2012. – Vol. 933. – P. 161-74. doi: 10.1007/978-1-62703-068-7_10.

Point 2: I know some researchers have published articles before.

Response 2: We do not quite understand the meaning of this comment. There are a lot of articles on extra-islet insulin-producing cells, then, of course, this new area of research is being intensively developed. This cells, as well as issues related to the mechanisms of their formation, undoubtedly represent both important practical (restoration of the insulin-producing function of the pancreas in diabetes) and fundamental theoretical interest (study of the regulatory systems of the body in normal and pathological conditions, functional plasticity of cells). We can say that on the basis of the laboratory of Morphology and Biochemistry of the Institute of Immunology and Physiology (Ural Branch of the Russian Academy of Sciences), a number of works were carried out to study the action of APH in vivo and in vitro, which were reflected in publications (Danilova et al., 2017 - number 34 in the reference list; Pozdina et al., 2021 - number 33; Danilova, IG, Shafigullina, ZA, Gette, IF, Sencov, VG, Medvedeva, SY, & Abidov, MT (2020) Accelerated liver recovery after acute CCl4 poisoning in rats treated with sodium phthalhydrazide International Immunopharmacology, 80, [106124]. https://doi.org/10.1016).

The experience of using APH to stimulate the formation of extra-islet insulin-producing cells of the pancreas in experimental type 2 diabetes was highlighted at conferences (30th Congress of the ESP, 14th World Congress on Endocrinology and Diabetes, USBEREIT 2019, 32nd Congress of the ESP) and publications based on conference materials. Full-scale articles describing the effect of APH on the formation of extra islet IPC are not known to us.

Point 3: The mechanistic part of APH is completely lacking in the manuscript.

Response 3: In works [30-34], it was shown that AFG contributes to a change in the production of macrophages towards anti-inflammatory. In [34], in a rat model of alloxan diabetes, it was shown decreasing macrophage infiltration in alloxan-damaged pancreas and reprogramming the remaining macrophages to produce specific signals, possibly required for β-cell growth. In our work, we used AFG to demonstrate how a decrease in the number of pro-inflammatory macrophages in the pancreas and an increase in the production of the anti-inflammatory growth factor TGF-β1 can affect the functional activity of pancreatic cells using extra-islet insulin-producing cells as an example.

Point 4: The authors should include the plasma concentration of 2-methoxyestradiol and catechol-o-methyl-transferase levels in the treatment group.

Response 4: 2-Methoxyestradiol or resveratrol has antioxidant properties. In our previous work, we studied the effect of isoflavones, which are also antioxidants, on the beta cells of rats with induced diabetes, these materials are presented in [Duru K.C. et al., 2020]. Materials on the effect of antioxidants on extra-islet IPC are being prepared for publication.

Duru, K.C., Mukhlynina, E.A., Moroz, G.A., Gette, I.F., Danilova, I.G., & Kovaleva, E.G. (2020). Anti-diabetic effect of isoflavone rich kudzu root extract in experimentally induced diabetic rats. Journal of Functional Foods, 68, [103922]. https://doi.org/10.1016/j.jff.2020.103922

 Catechol-o-methyltransferase (COMT) is an enzyme catalyzing the inactivation of catecholamines (dopamine, norepinephrine, epinephrine) by attaching a methyl group. Adrenaline and norepinephrine are contrainsular hormones; in diabetes mellitus, their increased formation, along with glucagon and glucocorticoids, accelerates gluconeogenesis. Discovered by Wang J.P. et al., 2002 the decrease in COMT activity in the liver and plasma of rats with streptozotocin diabetes mellitus (type 1 DM model) may be associated with destructive processes in the liver or with a compensatory response to enhance gluconeogenesis. In type 2 diabetes mellitus, gluconeogenesis is less pronounced than in type 1 diabetes. According to the literature, some genetic variants of COMT are associated with different accumulation of glycated hemoglobin in T2DM (Hall KT et al., 2016), difficulty in treating the disease (Bozek T. et al., 2017), and an increased risk of cardiovascular complications (Hall KT et al., 2017). al., 2016). The duration of action of catecholamines, depending on the rate of their inactivation, in various genetic variants of COMT, obviously, should affect the severity of gluconeogenesis, the state of the myocardium, catabolic reactions, and, accordingly, the weight of patients. The rats used in our experiment belonged to the same Wistar genetic line and were not genetically heterogeneous, we modeled type 2 diabetes, so we did not plan to determine the activity of COMT.

As a contrainsular hormone, we determined the content of corticosterone in plasma. We did not get significant differences between groups, but added this parameter to the manuscript to clarify this issue (lines 118-120 and 206-207).

Hall KT, Jablonski KA, Chen L, Harden M, Tolkin BR, Kaptchuk TJ, Bray GA, Ridker PM, Florez JC; Diabetes Prevention Program Research Group, Mukamal KJ, Chasman DI. Catechol-O-methyltransferase association with hemoglobin A1c. Metabolism. 2016 Jul;65(7):961-967. doi: 10.1016/j.metabol.2016.04.001. Epub 2016 Apr 14. PMID: 27282867; PMCID: PMC4924514.

COMT rs4680 high-activity G-allele was associated with lower HbA1c and modest protection from type 2 diabetes. The directionality of COMT associations was concordant with those previously observed for cardiometabolic risk factors and CVD.

Bozek T, Blazekovic A, Perkovic MN, Jercic KG, Sustar A, Smircic-Duvnjak L, Outeiro TF, Pivac N, Borovecki F. The influence of dopamine-beta-hydroxylase and catechol O-methyltransferase gene polymorphism on the efficacy of insulin detemir therapy in patients with type 2 diabetes mellitus. Diabetol Metab Syndr. 2017 Dec 4;9:97. doi: 10.1186/s13098-017-0295-0. PMID: 29225702; PMCID: PMC5716004.

Wang JP, Liu IM, Tzeng TF, Cheng JT. Decrease in catechol-O-methyltransferase activity in the liver of streptozotocin-induced diabetic rats. Clin Exp Pharmacol Physiol. 2002 May-Jun;29(5-6):419-22. doi: 10.1046/j.1440-1681.2002.03673.x. PMID: 12010186. 

Point 5: What is the phosphorylation levels of AMPK in these extra-Islet Insulin Producing cells treated with APH?

Response 5: AMP activated protein kinase (AMPK) controls the energy balance of the cell; it is activated when the cell consumes significant energy and increases the intracellular level of AMP. As a result of AMPK activation in the cell, the synthesis of fatty acids decreases and their oxidation increases. Increased fatty acid oxidation with accumulation of ketone bodies is characteristic of type 1 diabetes mellitus with high glucose levels. There are in vivo experimental data on increased apoptosis of β-cells when AMPK is activated in them (Kefas B.A. et al., 2004), probably due to the depletion of energy resources. In another in vitro study, on the contrary, a dose-dependent decrease in β-cell apoptosis was noted during AMPK activation, probably due to a decrease in the excess amount of triglycerides (Nyblom H.K. et al., 2008). The study of energy metabolism in T2DM is certainly an interesting topic, but in this experiment, we did not plan to study energy metabolism in diabetes in the body as a whole and in β-cells. The glucose content in our T2DM model was moderate. It is possible that AMPK activation and increased lipolysis could reduce insulin resistance in obese rodents, but we did not use a fat diet in our model.

Kefas BA, Cai Y, Kerckhofs K, Ling Z, Martens G, Heimberg H, Pipeleers D, Van de Casteele M. Metformin-induced stimulation of AMP-activated protein kinase in beta-cells impairs their glucose responsiveness and can lead to apoptosis. Biochem Pharmacol. 2004 Aug 1;68(3):409-16. doi: 10.1016/j.bcp.2004.04.003. PMID: 15242807.

Nyblom HK, Sargsyan E, Bergsten P. AMP-activated protein kinase agonist dose dependently improves function and reduces apoptosis in glucotoxic beta-cells without changing triglyceride levels. J Mol Endocrinol. 2008 Sep;41(3):187-94. doi: 10.1677/JME-08-0006.

Point 6: Include one positive control group in your studies such as metformin. 

Response 6: We were not faced with the task of comparing the effectiveness of treating T2DM with different drugs. The purpose of this study was driven by the desire to contribute to knowledge about the extent of macrophages' contribution to functional variability of adult pancreatic cells during T2DM progression. As a tool for influencing macrophages, we used APH, the ability of which to shift the production of macrophages towards anti-inflammatory was shown earlier [30-34], and the extra-island insulin-producing cells of the pancreas became the object of study. As for metformin, its mechanism of action is associated with the ability to suppress gluconeogenesis, the formation of free fatty acids and fat oxidation. Metformin increases the sensitivity of peripheral receptors to insulin and the utilization of glucose by cells, but it does not affect the amount of insulin in the blood and its formation in the pancreas.

Round 2

Reviewer 1 Report

All the suggestions and comments are addressed appropriately and comprehensively. Thank you!